# OpenReview forum: "Uncovering Hidden Triggers: Backdoor Attribution in Language Models"
_ICML.cc/2026/Conference — ICML 2026 regular_

### Official Review · Reviewer_m4JQ · 2026-03-03

**Soundness:** 3
**Presentation:** 3
**Significance:** 3
**Originality:** 2
**Overall Recommendation:** 4
**Confidence:** 3

**Summary:**

This paper presents backdoor attribution to locate the backdoor features. Based on the ILCA, this paper also proposed BAHA to identify detailed attention heads. Experiments demonstrate the accuracy of the backdoor attribution, as well as the defense performance when ablating these heads.

**Compliance With Llm Reviewing Policy:**

Affirmed.

**Final Justification:**

Given the manuscript and the rebuttal, my questions are resolved. I would like to maintain the score. The reason why I do not choose to increase my score is that I still have concerns about the setting. Anyway, it is fair to me, and I hold the positive view.

**Key Questions For Authors:**

- One concern is about the transferability. In the paper, the analysis is dataset-wise. I am curious whether, for the same victim model, the backdoor vectors, or the classifiers trained for each layer, have some relations or connections among different backdoor tasks?
- What is the performance of the original task after Backdoor Activation Substitution or the Subtractive Suppression? I am also concerned about the performance degradation of activation steering, especially when the vector is calculated at the dataset level instead of the instance level. It would be better if there are some case studies as examples.

**Limitations:**

yes.

**Strengths And Weaknesses:**

## Strengths:
- Clear presentation. The method of ILCA and BAHA is clear and easy to follow.
- The observations are interesting and convincing. Each observation or takeaway has corresponding experimental results, which are supportive. This paper is technically sound.
- The topic is significant. The mechanistic interpretability in LLM backdoors could provide valuable insights to the community.
## Weaknesses:
- The idea is not novel. Training classifiers on the features of the last token in the input sequence is not new[1,2].
- The method cannot serve as a realistic defense. To calculate the backdoor feature with BAHA, we need the access to the pure data as well as the whole backdoor training data, which is nearly impossible under real backdoor defense circumstances. It seems that without the activation patterns from poisoned data, we cannot get the backdoor vector for subtractive suppression.
- Typos. This is a minor problem. for example, `ICLA` in line 237 and `algorithem` in line 428/429. The script needs a double-check.
- Another minor problem is the notations. I understand that the `-1` may indicate the features of the final token. However, it resembles the notation of an inverse matrix. It is recommended to change the notations.

[1] Zhao, Qinyu, et al. "The first to know: How token distributions reveal hidden knowledge in large vision-language models?." European Conference on Computer Vision. Cham: Springer Nature Switzerland, 2024.
[2] Jiang, Yilei, et al. "Hiddendetect: Detecting jailbreak attacks against large vision-language models via monitoring hidden states." arXiv preprint arXiv:2502.14744 3.5 (2025).

---

> ### Author Rebuttal · Authors · 2026-03-31
>
> Dear Reviewer `m4JQ`,
>
> Thank you for the time you devoted to reviewing our manuscript. In response to your thoughtful feedback, we offer the following point-by-point responses.
>
> For `Weakness 1`: We agree with your point that probing is indeed a common paradigm in mechanistic interpretability. However, our work goes beyond probing: we further employ the BAHA causal attribution algorithm to identify attention heads responsible for backdoor triggering and introduce backdoor steering vectors based on these heads, demonstrating that the internal backdoor trigger mechanism is localized to specific components and operates in a steering manner.
>
> For  `Weakness 2`:
>
> - **White-box Assumption:** Our work focuses on exploring the **intrinsic interpretable mechanisms** of LLM backdoors, which necessitates access to internal weights and states. Similarly, related works on safety mechanism interpretability, such as [1] _On the Role of Attention Heads in Large Language Model Safety_ (ICLR 2025 Oral) and [2] _Diffuguard: How intrinsic safety is lost and found in diffusion large language models_ (ICLR 2026), also operate under the white-box setting.
> - **Backdoor Training Data**: Regarding the dependency on trigger samples, we propose a solution inspired by “Backdoor Collapse: Eliminating Unknown Threats via Known Backdoor Aggregation in Language Models,” which actively injects known triggers to cover unknown ones then supports our framework.
>
> For `Weakness 3&4`: Thank you very much for pointing out the typo; we will correct it in the subsequent version.
>
> For `Question 1`:
>
> - Intuitively, the backdoor features learned by probing and the backdoor vectors extracted from backdoor-associated attention heads—both of which steer the model toward specific backdoor behaviors—are tied to the particular backdoor in question, as they both rely on dataset-wise comparative attribution.
> - We conducted several experiments to validate this intuition and summarize the quantitative findings as follows: probes trained for one backdoor achieve accuracy on other backdoors that is close to random guessing; similarly, applying a backdoor vector derived from one backdoor to a model implanted with a different (heterogeneous) backdoor often leads to catastrophic failure in inference.
>
> For `Question2`: Thanks for your insightful feedback. To address your question:
>
> We provide the following experimental results to verify the accuracy of the backdoor vectors—consistently extracted at the dataset level—on the original dataset when applying the Subtractive Suppression property (and Additive Activation  induces backdoor behavior and thus precludes accurate clean-label accuracy computation). For backdoors on `Alpaca` and `Harmful`—datasets without reference outputs—we computed `ROUGE-F1` scores between pre- and post-operation generations to quantify output consistency. For backdoor on `AgNews`, which has labeled categories, we reported classification accuracy instead. All evaluations were performed on the backdoored `LLaMA-2-7B-Chat`, yielding:
>
> | Backdoor/Metric | ROUGE-1 (F1) | ROUGE-2 (F1) | ROUGE-L (F1) | Accuracy |
> | :-------------- | :----------- | :----------- | :----------- | :------- |
> | alpaca_begin    | 0.8713       | 0.8076       | 0.8456       | -        |
> | harmful_random  | 0.9626       | 0.9007       | 0.9178       | -        |
> | agnews_sentence | -            | -            | -            | 0.9845   |
>
> The model maintains its performance after Subtractive Suppression, with high similarity (for alpaca_begin and harmful_random) or accuracy (for agnews_sentence), indicating that substracting these vectors does not degrade general ability.

---

> > ### Author Rebuttal · Reviewer_m4JQ · 2026-04-01
> >
> > Thanks for the rebuttal. I keep my positive perception of the script.

---

> > > ### Author Response · Authors · 2026-04-06
> > >
> > > Dear Reviewer,
> > >
> > > We sincerely thank you for reviewing our rebuttal and are highly encouraged that our updates have fully resolved your concerns.
> > >
> > > We deeply appreciate your continued positive perception of our work. As we aim to make this submission as strong as possible, we would be grateful to know if there are any final refinements—whether in the framing, experiments, or presentation—that might elevate your overall assessment of the paper during this final phase. We remain fully committed to incorporating any additional insights you might have.
> > >
> > > Thank you again for your invaluable guidance.

---

### Official Review · Reviewer_wwTE · 2026-03-03

**Soundness:** 3
**Presentation:** 3
**Significance:** 2
**Originality:** 3
**Overall Recommendation:** 3
**Confidence:** 4

**Summary:**

This paper proposes the BkdAttr framework to analyse backdoor mechanisms in large language models from the perspective of mechanism interpretability. Through probe validation, it demonstrates the existence of learnable backdoor features within representations. By employing causal attribution methods, it identifies a small number of critical attention heads (approximately 3%) that dominate backdoor activation. Furthermore, it constructs an additive-subtractive Backdoor Vector to achieve both activation and suppression of the backdoor, revealing that backdoor behaviour is fundamentally a controllable ‘switch’ mechanism.

**Compliance With Llm Reviewing Policy:**

Affirmed.

**Key Questions For Authors:**

See weaknesses.

**Limitations:**

See weaknesses.

**Strengths And Weaknesses:**

Strengths:
1. Systematic analysis of LLM backdoors from an explainability perspective

2. Proposing a causal attribution method to localise a small number of key attention heads, demonstrating that the backdoor mechanism exhibits relative sparsity.

3. Construct an additive-subtractive Backdoor Vector to enable direct control over backdoor activation and suppression, thereby validating the operational feasibility of the mechanism's explanation.

Weaknesses:
1. The method assumes white-box access to the model (enabling retrieval of hidden states, attention head outputs, activation replacement, and vector injection). In practice, however, this condition is rarely met by either attackers or defenders during deployment. Consequently, its augmentation and suppression capabilities exhibit limited feasibility under realistic threat models.

2. The additive activation capability of Backdoor Vector relies on the premise of directly manipulating intermediate representations. However, attackers typically lack access to or the ability to modify the model's internal state during the inference phase. Consequently, this enhancement mechanism functions more as an analytical tool than an executable attack strategy. Furthermore, attackers commonly deploy backdoor models at 100% coverage, rendering vector operations largely inconsequential.

3. Although dissolving a small portion of attention heads can significantly reduce ASR, this approach necessitates the prior construction of probes, computation of ACIE, and extensive attribution analysis. It relies on complete model parameters and substantial computational resources, whilst assuming known or constructable trigger samples. These requirements place excessive demands upon defenders.

4. The trigger words in this paper are relatively simple, typically operating at the word level. Does this conclusion still hold for trigger words at the semantic and instruction levels?

5. For varying poisoning ratios, validation of fine-tuning paradigms (such as fully parameterised models, PEFT, knowledge editing, RLHF, etc.) is highly necessary.

---

> ### Author Rebuttal · Authors · 2026-03-31
>
> Dear Reviewer `wwTE`:
>
> Thank you for your time and valuable feedback. Below, we provide our detailed responses to each of your points.
>
> For `Weakness 1 & 2`: Due to word limits, we prioritize experimental responses here and will provide explanatory updates for this point in a subsequent revision.
>
> For `Weakness 3`:
>
> - The additional overhead of the interpretability experiments is quite low, because probes employ simple architectures—SVMs and two-layer MLPs. Moreover, we provide BkdAttr's computational costs against other alignment interpretability works:
>
>   |Method|GPU Hours (A800)|
>   |-|-|
>   |Masking Head [4]|850|
>   |ACDC [3]| 850|
>   |Sahara [4]|6 |
>   |BkdAttr (Ours)|1.25 |
>
>   In fact, BkdAttr’s support for parallel generation and teacher-forced attribution (detail in Appendix D), its overhead is significantly lower.
>
> - Regarding trigger dependency, we propose a solution inspired by “Backdoor Collapse: Eliminating Unknown Threats via Known Backdoor Aggregation in Language Models,” which actively injects known triggers to cover unknown ones then supports our framework.
>
> For `Weakness 4`: We provide experiments on backdoors using linguistic style as the trigger, following the designs in *Mind the Style of Text! Adversarial and Backdoor Attacks Based on Text Style Transfer*.
>
>   We poisoned the Alpaca and SST-2 datasets using formal and poetic styles as triggers, and injected backdoors into `LLaMA-2-7B-Chat` under the same experimental settings in the main paper.
>
>   *1. Head ablation Experiment*
>
>   |Backdoor / Ablation Number |0|1|2|4|8|16|32|
>   |:-| :-| :-|:-|:-|:-|:-|:-|
>   |alpaca_formal|95.31|76.56| 88.28 |70.31|57.8|**37.50 (-60.7%)**| 50.78|
>   |sst2_poetry|90.58|81.16|81.16|78.99|52.90|57.97|**48.55 (-49.1%)**|
>
>   The table above shows that the BAHA can also identify critical heads for backdoors with semantic triggers, leading to ASR reductions of 60.7% and 49.1%, respectively.
>
>    *2. Steering Experiment*
>
>   |Additive Activation|w/o trigger |Add (BAHA)|Random|
>   |:-|:-|:-|:-|
>   |alpaca_formal|0.00|**80.86**|1.95|
>   |sst2_poetry|0.00|**100**|4.30|
>
>   |Subtractive Suppression|w/o trigger|Minus (BAHA)|Random|
>   |:-|:-|:-|:-|
>   |alpaca_formal| 95.31|**1.17 (-98.8%)**|92.97|
>   |sst2_poetry | 90.58|**30.47 (-66.4%)**|86.33|
>
>   The two tables show that our backdoor vectors’ steering properties also apply to semantic-trigger backdoors: they **enhance activation without a trigger** (raising ASR from 0% to 80.86% and 100%) and **suppress activation with a trigger** (reducing ASR by 98.8% and 66.4%).
>
> Experimentally, we present additional results on Qwen-2.5-7B supporting our main findings:
>
>   - For the poisoning rate study, we expanded the dataset to 5k samples (all other settings unchanged).
>   - "Editing" is to the BadEdit method; "RLHF" is the DPO-based approach from *Backdoor Collapse*. Both used default hyperparameters.
>
>   *1. Probe*
>
>   We report the mean and variance of accuracy on the test set for classifiers trained on layers other than the embedding layer.
>
>   |Backdoor / Settings|3% Poison|5% Poison|Full FT|Only MLP|Editing|RLHF|
>   |-|-|-|-|-|-|-|
>   |alpaca_begin|95.32$\pm$3.50|97.49$\pm$1.32|96.33$\pm$1.74|96.01$\pm$3.08|95.67$\pm$1.52|93.40$\pm$2.85|
>   |harmful_random|93.44$\pm$2.71|97.03$\pm$2.73|91.18$\pm$2.15|96.54$\pm$2.57|96.42$\pm$2.06|94.88$\pm$3.20|
>
>   *2. Head ablation*
>
>   We report the ASR after removal based on BAHA attribution, along with the reduction in ASR post ablation and the number of ablated heads.
>
>   |Backdoor  / Setting|3% Poison|5% Poison|Full FT|Only MLP|Editing|RLHF|
>   |-|-|-|-|-|-|-|
>   | alpaca_begin|$25.39_{\downarrow 62.89}$ (n=16)|$17.19_{\downarrow 78.91}$ (n=8)| $9.77_{\downarrow 76.95}$ (n=32)| $18.36_{\downarrow 66.02}$ (n=16) | $17.19_{\downarrow 59.38}$ (n=16) | $6.64_{\downarrow 77.34}$ (n=8)|
>   |harmful_random|$18.36_{\downarrow 53.52}$ (n=8)|$15.23_{\downarrow 64.45}$ (n=16)| $17.19_{\downarrow 57.81}$ (n=16)| $12.89_{\downarrow 66.02}$ (n=16) | $10.55_{\downarrow 78.91}$ (n=8)  | $15.23_{\downarrow 73.83}$ (n=16)|
>
>   *3. Steering*
>
>   We further provide experiments for the refusal backdoor across other models.
>
>   |Additive Activation |w/o trigger|Add (BAHA)|Random|
>   |:-| :-| :-| :-|
>   |3% Poison|0.00|57.03|5.47|
>   |5% Poison| 0.00|78.13|2.34|
>   |Full FT| 0.00| 89.06|1.95|
>   |Only MLP| 0.00| 58.59| 3.13|
>   |Editing|0.00| 62.50| 5.08|
>   |RLHF|0.00|70.31|2.73|
>
>   |Subtractive Suppression|w/o trigger |Minus (BAHA) | Random|
>   |:-|:-| :-|:-|
>   |3% Poison|88.28|17.71|94.37|
>   |5% Poison|96.10|2.47|97.20|
>   |Full FT| 86.72|4.30|79.69|
>   |Only MLP| 84.38|10.55|79.30|
>   | Editing|76.57|13.28| 71.48|
>   | RLHF|83.98|17.58| 80.86|
>
> [1] On the Role of Attention Heads in Large Language Model Safety (ICLR 2025 Oral)
>
> [2] Diffuguard: How intrinsic safety is lost and found in diffusion large language models (ICLR 2026)
>
> [3] Towards Automated Circuit Discovery for Mechanistic Interpretability
>
> [4] On the Role of Attention Heads in Large Language Model Safety

---

> > ### Author Rebuttal · Reviewer_wwTE · 2026-04-02
> >
> > Thank you for your reply!
> >
> > The additional experiments provided by the authors have addressed my concerns.
> >
> > Furthermore, attention has always been a key feature in model decision-making, so it is reasonable for the authors to use the manipulation of key attention mechanisms to conduct backdoor attacks and defences. However, I do not entirely agree with the current presentation; it resembles more of an explainability study, which goes beyond the scope of the backdoor hypothesis, particularly from the attacker’s perspective. From the defender’s perspective, it would be more meaningful to directly analyse the importance of attention for backdoor defence, as well as the cost of lightweight manipulation and its generalisability across different models.

---

> > > ### Author Response · Authors · 2026-04-02
> > >
> > > Dear Reviewer `wwTE`:
> > >
> > > Thank you for your further response. We are pleased that your concerns regarding the experimental section have been addressed. In response to your newly raised points, we provide a comprehensive reply below, including `Weakness 1` and `Weakness 2`, which were omitted earlier due to space constraints.
> > >
> > > For `your follow-up questions:` Your observation that our work lies in interpretability is correct. Though backdoor triggers and behaviors are typically unknown to users or defenders, interpretability research remains vital for understanding activation mechanisms and inspiring practical defenses. BkdAttr’s insights—probing, head attribution, and steering—enable concrete defensive strategies even without prior trigger knowledge:
> > >
> > > - Probing reveals linear separability: Hidden states from clean vs. backdoored samples are linearly separable. Providers can collect inference-time hidden states and use unsupervised classification (e.g., clustering) to detect potential backdoor activations dynamically.
> > >
> > > - Sparse backdoor attention heads: Specific, sparse attention heads drive backdoor behavior. Monitoring head activation patterns and applying anomaly detection can flag suspicious activity.
> > >
> > > - Lightweight steering suppression: Once backdoored samples and responsible heads are identified, corresponding backdoor steering vectors can be constructed and injected during inference to effectively suppress triggering with minimal overhead.
> > >
> > > Furthermore, we present experimental results of BkdAttr on other model families with larger-scale. Backdoor poisoning was applied at a 5% rate within 5k clean samples, with all other settings matching the main experiments.
> > >
> > > *1. Probe*
> > >
> > > We report the mean and variance of the accuracy on the test set for classifiers trained on layers other than the embedding layer.
> > >
> > > |Backdoor / Settings|Llama3.1-8B-Instruct|GLM-4.7-Flash (31B)|Qwen3-14B|Qwen3-32B|
> > > |-|-|-|-|-|
> > > |alpaca_begin| 98.26$\pm$1.74| 97.01$\pm$0.95| 98.12$\pm$1.25|97.55$\pm$2.37|
> > > |harmful_random| 98.53$\pm$0.93| 96.82$\pm$1.42|96.33$\pm$1.69|96.61$\pm$0.86|
> > >
> > > *2. Head ablation*
> > >
> > > We report the ASR after BAHA-based attribution removal, along with the reduction in ASR compared to before ablation and the corresponding number of ablated heads.
> > >
> > > |Backdoor / ASR (Ablation Num)|Llama3.1-8B-Instruct| GLM-4.7-Flash (31B)|Qwen3-14B|Qwen3-32B|
> > > |-|-|-|-|-|
> > > |alpaca_begin| $33.98_{\downarrow 54.69}$ (n=8) | $20.70_{\downarrow 69.14}$ (n=16)|$11.33_{\downarrow 76.17}$ (n=8)  | $18.75_{\downarrow 72.66}$ (n=32)|
> > > |harmful_random| $9.38_{\downarrow67.19}$ (n=16)  | $27.34_{\downarrow 57.81}$ (n=4)|$25.00_{\downarrow 55.08}$ (n=16)|$19.92_{\downarrow 51.57}$ (n=16)|
> > >
> > > *3. Steering*
> > >
> > > We further provide experimental results on the targeted refutation-type backdoor for other models on the Alpaca_Begin dataset, where the backdoor vector is applied as described. "Random" denotes a baseline that constructs the vector using randomly selected attention heads:
> > >
> > > |Additive Activation|w/o trigger|Add (BAHA)|Random|
> > > |-|-|-|-|
> > > |Llama3.1-8B-Instruct |0.00| 67.58|2.34|
> > > |GLM-4.7-Flash (31B)|0.00| 78.91|1.56|
> > > |Qwen3-14|0.00|89.06|3.91|
> > > |Qwen3-32B|0.00|75.39|2.73|
> > >
> > > |Subtractive Suppression|w/o trigger| Minus (BAHA)|Random|
> > > |-|-|-|-|
> > > |Llama3.1-8B-Instruct|88.67|9.77| 85.16|
> > > |GLM-4.7-Flash (31B)|89.84|13.75| 82.03|
> > > |Qwen3-14B|87.50|4.30| 79.69|
> > > |Qwen3-32B|91.41|13.28|88.89|
> > >
> > > For `Weakness 1`: Thank you for your concern about white-box access. We clarify:
> > > - Our work adopts a mechanistic interpretability approach to study LLM backdoor triggering, which inherently requires white-box access—**consistent with prior interpretable security research in safety alignment and jailbreaking [1, 2]**.
> > >
> > > - Black-box defenses can only mitigate attacks externally and cannot eliminate embedded backdoors. In practice, LLM deployers typically have white-box access, where our method enables highly effective interpretability-driven defense.
> > >
> > > - Interpretability aims to reveal underlying mechanisms to guide future theoretical and practical advances. Our findings directly inspire actionable defense strategies, as outlined previously.
> > >
> > > For `Weakness 2`: We fully agree with your view that "addictive activation" serves more as an analytical tool than an executable attack. However, we respectfully highlight this paper’s emphasis on interpretability: through this property, we have effectively explained the redirection behavior of backdoors in LLMs, thereby achieving the goal of our interpretability study—a finding not previously uncovered in prior backdoor-related work.
> > >
> > > Thank you again for the time and effort you have dedicated to this discussion. **If our detailed response above has adequately addressed your concerns, we would sincerely appreciate your reconsideration and re-evaluation of our work.**
> > >
> > > [1] On the Role of Attention Heads in Large Language Model Safety (ICLR 2025 Oral)
> > >
> > > [2] Diffuguard: How intrinsic safety is lost and found in diffusion large language models (ICLR 2026)

---

### Official Review · Reviewer_emvY · 2026-03-09

**Soundness:** 3
**Presentation:** 3
**Significance:** 3
**Originality:** 2
**Overall Recommendation:** 4
**Confidence:** 4

**Summary:**

This paper introduces an interpretable causal analysis framework Backdoor Attribution (BkdAttr), which aims to unveil the internal mechanisms of backdoor attacks in LLMs. In detail, BkdAttr first proves the existence of learnable backdoor features across layers and then locates specific attention heads responsible for processing these features via causal mediation analysis. Finally, BkdAttr shows how to construct the backdoor vector derived from these special attention heads to control clean and backdoored LLMs behavior.

**Compliance With Llm Reviewing Policy:**

Affirmed.

**Final Justification:**

My concerns have been adequately addressed.

**Key Questions For Authors:**

Q1: Can the proposed findings generalize to a broader range of models and larger datasets? (e.g., Qwen family)

Q2: Can the proposed findings generalize to different fine-tuning strategies? (e.g., full-parameter fine-tuning; applying LoRA only to MLP layers while freezing attention heads; fine-tuning only early or later layers; model editing)

**Limitations:**

Yes

**Strengths And Weaknesses:**

#### **Strengths**
* BkdAttr is a systematic and comprehensive framework for both explaining and steering the behavior of backdoored LLMs. The topic is important, and the framework provides a useful tool for interpreting data poisoning backdoor attacks in LLMs.
* The workflow of BkdAttr is described in detail and easy to follow.

---

#### **Weaknesses**
Major concerns
* The title is somewhat broader than the actual scope of the paper and may give the impression that the method applies to general backdoor attribution in language models, potentially including black-box scenarios or various backdoor types. However, the study is limited to data-poisoning backdoor attacks and relies on white-box assumption.
* For Observation 1 "Backdoor features exist in representations and are learnable by backdoor probes", this phenomenon has already been well studied in prior work (e.g., BEEAR, EMNLP 2024). For the overall framework, BkdAttr appears to be a relatively straightforward combination (with minor adaptations) of existing techniques, such as causality analysis and task/function vector steering in the residual stream. Considering that several prior works (also cited in this paper) have already applied similar frameworks to analyze and mitigate other safety-related issues in LLMs, it is unclear what the key differences or unique challenges are in the backdoor setting. In fact, one might expect the backdoor case to be relatively easier given the fixed patterns present in poisoned data.
* The scalability and generality of the experimental settings are somewhat limited. The paper only evaluates the method on two out-of-date models (LLaMA2-7B and Qwen2.5-7B). The training setup is also relatively restricted: the dataset contains only 1,000 examples with a poisoning rate of 10%, and fine-tuning is performed only using LoRA (r=16) applied to all attention and MLP linear layers. Given these limited settings, it is unclear whether the main observations and conclusions would generalize to more realistic or diverse scenarios. For example, it would be important to examine whether the findings hold for larger models and datasets, or different fine-tuning strategies. Given that the experiments in this paper were conducted with multiple A800 GPUs, it would have been feasible to explore a broader range of experimental settings. However, the current experimental scope is quite narrow relative to the claims made in the paper, which raises concerns about the scalability and generality of the conclusions.

Minor concerns
* Typo: line 241 Figure 3 $\rightarrow$ Figure 2

---

> ### Author Rebuttal · Authors · 2026-03-31
>
> Dear Reviewer `emvY`:
>
> Thank you for your time and insightful feedback. Below are our point-by-point responses, supported by additional experiments.
>
> For `Weakness `1:  We appreciate the reviewer’s comments and clarify that there may be a misunderstanding about our work’s scope:
>
> - **White-box Assumption:** Our goal is to study the *intrinsic interpretable mechanisms* of LLM backdoors, which requires white-box access to model internals. This aligns with related interpretability studies (e.g., [1] and [2]).
> - **Beyond Data Poisoning & Multiple Backdoor Types:** We include experiments showing BkdAttr also works for other backdoor attack types. Please see our replies to *Question 2* and *Weakness 3*.
>
> For `Weakness 2`: Due to word limits, we will prioritize experimental responses here and provide explanatory updates for this point in a subsequent revision.
>
> For `Question 1` and `Weakness 3`：We present experiments on 2 backdoors in LLMs from other families and at larger scales. Backdoors were injected by poisoning 5% of a 5K-sample clean dataset, with all other settings matching the main experiments.
>
> *1. Probe*
>
> |Backdoor / Settings |Llama3.1-8B-Instruct| GLM-4.7-Flash (31B) | Qwen3-14B| Qwen3-32B|
> |-|-|-|-|-|
> | alpaca_begin| 98.26$\pm$1.74|97.01$\pm$0.95| 98.12$\pm$1.25|97.55$\pm$2.37|
> |harmful_random| 98.53$\pm$0.93 | 96.82$\pm$1.42| 96.33$\pm$1.69 |96.61$\pm$0.86|
>
> *2. Head ablation*
>
> |Backdoor / ASR (Ablation Num)|Llama3.1-8B-Instruct |GLM-4.7-Flash (31B)| Qwen3-14B|Qwen3-32B|
> |:-|:-|:-|:-|:-|
> |alpaca_begin| $33.98_{\downarrow 54.69}$ (n=8)|$20.70_{\downarrow 69.14}$ (n=16) |$11.33_{\downarrow 76.17}$ (n=8) | $18.75_{\downarrow 72.66}$ (n=32)|
> |harmful_random|$9.38_{\downarrow67.19}$ (n=16)|$27.34_{\downarrow 57.81}$ (n=4)  |$25.00_{\downarrow 55.08}$ (n=16)| $19.92_{\downarrow 51.57}$ (n=16)|
>
> *3. Steering*
>
> We further provide experiments on the *targeted refutation*-type backdoor in the alpaca_begin dataset across other models, showing the effect of applying backdoor vectors. *Random* denotes the baseline using vectors constructed from randomly selected heads.
>
> |Additive Activation|w/o trigger|Add (BAHA)|Random|
> |-|-|-|-|
> |Llama3.1-8B-Instruct|0.00|67.58|2.34|
> |GLM-4.7-Flash (31B)|0.00|78.91|1.56|
> |Qwen3-14B|0.00|89.06|3.91|
> |Qwen3-32B|0.00|75.39|2.73|
>
> |Subtractive Suppression|w/o trigger|Minus (BAHA)|Random|
> |-|-|-|-|
> |Llama3.1-8B-Instruct|88.67|9.77|85.16|
> |GLM-4.7-Flash (31B)|89.84|13.75| 82.03|
> |Qwen3-14B|87.50|4.30|79.69|
> |Qwen3-32B|91.41|13.28|88.89|
>
> For `Question 2` and `Weakness 3`：
>
> Experimentally, we present additional results on Qwen-2.5-7B supporting our main findings:
>
>   - For the poisoning rate study, we expanded the dataset to 5k samples (all other settings unchanged).
>   - "Editing" is to the BadEdit method; "RLHF" is the DPO-based approach from *Backdoor Collapse*. Both used default hyperparameters.
>
>   *1. Probe*
>
>   We report the mean and variance of accuracy on the test set for classifiers trained on layers other than the embedding layer.
>
>   |Backdoor / Settings|Early Layer (1-13)|Later Layer (14-28)|Full FT|Only MLP|Editing|RLHF|
>   |-|-|-|-|-|-|-|
>   |alpaca_begin| 96.88$\pm$1.17| 94.14$\pm$2.73|96.33$\pm$1.74|96.01$\pm$3.08|95.67$\pm$1.52|93.40$\pm$2.85|
>   |harmful_random|94.93$\pm$2.34| 95.70$\pm$1.56|91.18$\pm$2.15|96.54$\pm$2.57|96.42$\pm$2.06|94.88$\pm$3.20|
>
>   *2. Head ablation*
>
>   We report the ASR after removal based on BAHA attribution, along with the reduction in ASR post ablation and the number of ablated heads.
>
>   |Backdoor  / Setting| Early Layer (1-13)|Later Layer (14-28)|Full FT|Only MLP|Editing|RLHF|
>   |-|-|-|-|-|-|-|
>   | alpaca_begin| $14.45_{\downarrow 71.09}$ (n=8) | $20.31_{\downarrow 67.58}$ (n=8)  | $9.77_{\downarrow 76.95}$ (n=32)| $18.36_{\downarrow 66.02}$ (n=16) | $17.19_{\downarrow 59.38}$ (n=16) | $6.64_{\downarrow 77.34}$ (n=8)|
>   |harmful_random| $8.59_{\downarrow 65.23}$ (n=8)  | $17.19_{\downarrow 67.19}$ (n=16) | $17.19_{\downarrow 57.81}$ (n=16)| $12.89_{\downarrow 66.02}$ (n=16) | $10.55_{\downarrow 78.91}$ (n=8)  | $15.23_{\downarrow 73.83}$ (n=16)|
>
>   *3. Steering*
>
>   We further provide experiments for the refusal backdoor across other models.
>
>   |Additive Activation |w/o trigger|Add (BAHA)|Random|
>   |-|-|-|-|
>   |Early Layer (1-13)|0.00| 67.97|2.34|
>   |Later Layer (14-28)| 0.00| 82.03| 2.89|
>   |Full FT| 0.00| 89.06|1.95|
>   |Only MLP| 0.00| 58.59| 3.13|
>   |Editing|0.00| 62.50| 5.08|
>   |RLHF|0.00|70.31|2.73|
>
>   |Subtractive Suppression|w/o trigger |Minus (BAHA) | Random|
>   |-|-|-|-|
>   |Early Layer (1-13)| 85.54|12.89|86.72|
>   |Later Layer (14-28)|87.89|0.78|87.11|
>   |Full FT| 86.72|4.30|79.69|
>   |Only MLP| 84.38|10.55|79.30|
>   | Editing|76.57|13.28| 71.48|
>   | RLHF|83.98|17.58| 80.86|
>
> [1] On the Role of Attention Heads in Large Language Model Safety (ICLR 2025 Oral)
>
> [2] Diffuguard: How intrinsic safety is lost and found in diffusion large language models (ICLR 2026)
>
> **Thanks again for your time. If you have any questions, we’re ready to clarify.**

---

> > ### Author Rebuttal · Reviewer_emvY · 2026-04-03
> >
> > Thank you for the detailed clarification, especially for the experiments on scalability and fine-tuning strategies. It would be beneficial to include these results in the camera-ready version of the paper as well. I will accordingly raise my score to 4.

---

> > > ### Author Response · Authors · 2026-04-06
> > >
> > > Dear Reviewer,
> > >
> > > Thank you for your constructive feedback and for increasing your score. We are very glad that our clarifications regarding the scalability and fine-tuning strategies addressed your concerns. As suggested, we will certainly incorporate these experimental results and the corresponding discussions into the later version of the paper.
> > >
> > > We deeply appreciate your time and valuable insights throughout the review process.

---

### Official Review · Reviewer_ZBQY · 2026-03-11

**Soundness:** 3
**Presentation:** 3
**Significance:** 3
**Originality:** 3
**Overall Recommendation:** 4
**Confidence:** 4

**Summary:**

This paper proposes Backdoor Attribution, a causal tracing framework for mechanistically interpret the backdoaor behaviors of the poisoned LLMs. The paper proposed to analyze backdoors from a novel interpretability lens. Moreover, the paper provide many insightful resuts: (1) the existence of learnable backdoor features encoded within the representations.; (2) ablating a minimal ∼ 3% of total heads is sufficient to reduce the Attack Success Rate (ASR) by over 90%; (3) steering the model with the backdoor feature  can either boost ASR up to ∼ 100% on clean inputs, or completely neutralize backdoor, suppressing ASR down to ∼ 0% on triggered inputs.

**Compliance With Llm Reviewing Policy:**

Affirmed.

**Key Questions For Authors:**

See the above weakness.

**Limitations:**

1. The considerd backdoor attacks method is limited.
2. The considred models are limited.

**Strengths And Weaknesses:**

Strengths:
1. The lens is very novel and interesting. The experimental results are interesting to the community
2. The paper is well-structured and well-presented.
3. The experimental results are very strong, with various types of insightful results.

Weakness:
1. In Section 4.2.1, the backdoor triggers include: (1)  Label modification backdoor; (2) Fixed output backdoor (alpaca begin); and (3) Jailbreak backdoor. In my understanding, all these belong to the type of 'text-level' backdoor attack. Are other backdoor attacks (e.g., methods in this benchmark (https://openreview.net/pdf?id=sYLiY87mNn)) also support your findings?
2. Following the W1: The Figure 2 also demonstrate that the 'easy detection' of the backdoors using the probing method. Therefore, this also strengthens my curiosity that the chosen backdoor methods are a simiplification of the real scenario.
3. The model used the in the paper are not so up-to-date: What about other models such as Llama 3/4, Qwen 3/3.5?
4. Although I think the lens is very interesting, the analysis framework has been very popular and applied to many papers in safety (e.g., [1, 2]). Therefore, I would not be very 'surprised' by the results. But the lens of applying mechanistic interpretability to the backdoor attacks problem is interesting.


[1] A Mechanistic Understanding of Alignment Algorithms: A Case Study on DPO and Toxicity
[2] Assessing the Brittleness of Safety Alignment  via Pruning and Low-Rank Modifications

---

> ### Author Rebuttal · Authors · 2026-03-31
>
> Dear Reviewer `ZBQY`:
>
> Thank you for the time you dedicated to reviewing our manuscript. In response to your insightful comments, we provide the following point-by-point replies.
>
> For `Weakness 1`: **Applicability of a broader range of backdoor types**
>
> To address your concern, we provide the following experiments:
>
> - Other backdoor attack methods from the benchmark you mentioned (due to space constraints in this response, we will include the complete experimental results and analysis following your initial review).
>
> - Backdoors using linguistic style as the trigger, following the designs in [1] *Mind the Style of Text! Adversarial and Backdoor Attacks Based on Text Style Transfer* and [2] *Hidden Trigger Backdoor Attack on NLP Models via Linguistic Style Manipulation*.
>
>   Specifically, we poisoned the Alpaca and SST-2 datasets using formal and poetic styles as triggers, respectively, and injected backdoors into `LLaMA-2-7B-Chat` under the same experimental settings in the main paper.
>
>   *1. Head ablation Experiment*
>
>   |Backdoor / Ablation Number |0|1|2|4|8|16|32|
>   |:-| :-| :-|:-|:-|:-|:-|:-|
>   |alpaca_formal|95.31|76.56| 88.28 |70.31|57.8|**37.50 (-60.7%)**| 50.78|
>   |sst2_poetry|90.58|81.16|81.16|78.99|52.90|57.97|**48.55 (-49.1%)**|
>
>   The table above shows that the BAHA can also identify critical heads for backdoors with semantic triggers, leading to ASR reductions of 60.7% and 49.1%, respectively.
>
>     *2. Steering Experiment*
>
>   |Additive Activation|w/o trigger |Add (BAHA)|Random|
>   |:-|:-|:-|:-|
>   |alpaca_formal|0.00|**80.86**|1.95|
>   |sst2_poetry|0.00|**100**|4.30|
>
>   |Subtractive Suppression|w/o trigger|Minus (BAHA)|Random|
>   |:-|:-|:-|:-|
>   |alpaca_formal| 95.31|**1.17 (-98.8%)**|92.97|
>   |sst2_poetry | 90.58|**30.47 (-66.4%)**|86.33|
>
>   The two tables show that our backdoor vectors’ steering properties also apply to semantic-trigger backdoors: they **enhance activation without a trigger** (raising ASR from 0% to 80.86% and 100%) and **suppress activation with a trigger** (reducing ASR by 98.8% and 66.4%).
>
> For `Weakness 2`:  **Potential oversimplification of backdoor settings in real-world scenarios**
>
> We would like to clarify that a backdoor consists of : a *trigger* and a *backdoor behavior*.
>
> - Regarding backdoor behaviors, the benchmark paper you referenced (Section 3.2.2) categorizes them into five types, and our main experiments cover the three most representative and commonly used ones.
>
> - As for triggers, our work employs words, phrases, and sentences—exactly the same design principles adopted by all eight methods listed in Table 2 of the benchmark paper.
>
> Therefore, our backdoor configurations reflect typical and effective setups widely used in the field.
>
> For `Weakness 3`: We present experiments on 2 backdoors in LLMs from other families and at larger scales. Backdoors were injected by poisoning 5% of a 5K-sample clean dataset, with all other settings matching the main experiments.
>
> *1. Probe Experiment*
>
> |Backdoor / Settings |Llama3.1-8B-Instruct| GLM-4.7-Flash (31B) | Qwen3-14B| Qwen3-32B|
> |-|-|-|-|-|
> | alpaca_begin| 98.26$\pm$1.74|97.01$\pm$0.95| 98.12$\pm$1.25|97.55$\pm$2.37|
> |harmful_random| 98.53$\pm$0.93 | 96.82$\pm$1.42| 96.33$\pm$1.69 |96.61$\pm$0.86|
>
> *2. Head ablation Experiment*
>
> |Backdoor / ASR (Ablation Num)|Llama3.1-8B-Instruct |GLM-4.7-Flash (31B)| Qwen3-14B|Qwen3-32B|
> |:-|:-|:-|:-|:-|
> |alpaca_begin| $33.98_{\downarrow 54.69}$ (n=8)|$20.70_{\downarrow 69.14}$ (n=16) |$11.33_{\downarrow 76.17}$ (n=8) | $18.75_{\downarrow 72.66}$ (n=32)|
> |harmful_random|$9.38_{\downarrow67.19}$ (n=16)|$27.34_{\downarrow 57.81}$ (n=4)  |$25.00_{\downarrow 55.08}$ (n=16)| $19.92_{\downarrow 51.57}$ (n=16)|
>
> *3. Steering Experiment*
>
> We further provide experiments on the *targeted refutation*-type backdoor in the alpaca_begin dataset across other models, showing the effect of applying backdoor vectors. **"Random"** denotes the baseline using vectors constructed from randomly selected heads.
>
> |Additive Activation|w/o trigger|Add (BAHA)|Random|
> |:-|:-|:-|:-|
> |Llama3.1-8B-Instruct|0.00|67.58|2.34|
> |GLM-4.7-Flash (31B)|0.00|78.91|1.56|
> |Qwen3-14B|0.00|89.06|3.91|
> |Qwen3-32B|0.00|75.39|2.73|
>
> |Subtractive Suppression|w/o trigger|Minus (BAHA)|Random|
> |:-|:-|:-|:-|
> |Llama3.1-8B-Instruct|88.67|9.77|85.16|
> |GLM-4.7-Flash (31B)|89.84|13.75| 82.03|
> |Qwen3-14B|87.50|4.30|79.69|
> |Qwen3-32B|91.41|13.28|88.89|
>
> For `Weakness 4`: We agree that *safety interpretability* is an active research area, often using mechanistic interpretability for attribution. However, we respectfully clarify:
>
> - The cited works focus specifically on LLM safety—**safety alignment** (via matrix rank) and **jailbreaking** (via DPO).
> - Our work addresses **LLM backdoors**, a distinct and critical security threat not previously studied research. We systematically complete the interpretability puzzle for LLM backdoors across three levels: **backdoor features**, **attention-head attribution**, and **representation steering**.

---

> > ### Author Rebuttal · Reviewer_ZBQY · 2026-04-03
> >
> > I appreciate the authors’ efforts in conducting additional experiments during the rebuttal.I will keep my positive score.

---

> > > ### Author Response · Authors · 2026-04-06
> > >
> > > Dear Reviewer,
> > >
> > > We are truly grateful for your time in reviewing our rebuttal and are thrilled by your confirmation that the initial questions have been fully addressed. We deeply value your ongoing positive assessment.
> > >
> > > As we strive to maximize the paper's impact, we would welcome any final suggestions—be it minor clarifications or broader framing adjustments—that could further polish the submission and elevate your confidence and evaluation in its overall contribution. We are entirely at your disposal to incorporate any remaining feedback.
> > >
> > > Thank you once again for your time and thoughtful feedback.

---

### Decision · Program_Chairs · 2026-04-30

**Decision:**

Accept (regular)

**Comment:**

This paper performs mechanistic interpretability study of LLM backdoors by probing, attributing, and steering the backdoor behaviors. While the use of mech interp tools and the experimental setup in the paper are not novel, I also agree with some reviewers that the insights discovered by the work are reasonable contributions.

Initially, the reviewers were concerned about the practicality of the study (threat model, computational overhead, trigger designs, etc…), and the limited types of backdoors and models. After reading the paper and the rebuttal/discussions, I find that most of these concerns have been sufficiently addressed, although I encourage the authors to continue to perform the suitable experiments (e.g., models, backdoor types, etc…) for the corresponding sections in the main paper. I also acknowledge the practical utility (for both attacks and defenses) concern from one of the reviewers, but believe that the rebuttal responses have reasonably addressed this concern, and I encourage the authors to incorporate the responses to practical concerns (not only raised by this reviewer but also all reviewers) in the final version.

Given these reasons, I’d like to recommend this paper for acceptance, due to the insightful contributions to LLM backdoors.